# Streams of conscious visual experience

Mar Martín-Signes ®[1,2] ✉, Ana B. Chica[1], Paolo Bartolomeo ®[3] & Michel Thiebaut de Schotten ®[2,4] ✉

Consciousness, a cornerstone of human cognition, is believed to arise from complex neural interactions. Traditional views have focused on localized fronto-parietal networks or broader inter-regional dynamics. In our study, we leverage advanced fMRI techniques, including the novel Functionnectome framework, to unravel the intricate relationship between brain circuits and functional activity shaping visual consciousness. Our findings underscore the importance of the superior longitudinal fasciculus within the fronto-parietal fibers, linking conscious perception with spatial neglect. Additionally, our data reveal the critical contribution of the temporo-parietal fibers and the splenium of the corpus callosum in connecting visual information with conscious representation and their verbalization. Central to these networks is the thalamus, posited as a conductor in synchronizing these interactive processes. Contrasting traditional fMRI analyses with the Functionnectome approach, our results emphasize the important explanatory power of interactive mechanisms over localized activations for visual consciousness. This research paves the way for a comprehensive understanding of consciousness, highlighting the complex network of neural connections that lead to awareness.

Consciousness comprises the fundamental aspect of sentient beings, including their perception, thoughts, and emotions[1,2]. In neuroscience, two prevailing theories attempt to elucidate its mechanisms. The Global Workspace Theory suggests that consciousness arises from the broadcasting of information within a global workspace, where sensory inputs compete for dominance after initial unconscious processing within primary modular cerebral networks[3]. In contrast, the Integrated Information Theory posits that consciousness emerges from a system's capacity to integrate information via the interactions between various regions of the brain, highlighting the crucial importance of neural networks[4–6]. Subjective conscious experiences in healthy individuals, whether conveyed through visual[7,8], somatosensory[9], or auditory[10] inputs, elicit distinctive local increases in functional activity within the frontal and parietal cortices[11]. However, the comprehensive understanding of the integrative mechanisms supporting the conscious experience remains elusive, partly due to the limitation of the traditional topological approach in functional magnetic resonance imaging (fMRI) studies[12]. These methods often overlook the dynamic interplay between brain regions, a critical component in the Integrated Information Theory framework[6,13]. To overcome these limitations, our study explores the role of anatomical connections in visual conscious experience using recent technical advancements like the Functionnectome[14–16].

The introduction of the Functionnectome represents a new methodology that integrates structural connectivity data within functional analysis. The Functionnectome takes the activity signals (blood oxygenation level-dependent [BOLD] time series) from the grey matter and combines them based on how these grey areas are connected to white matter areas. The strength of each connection influences the final signal. As a result, it creates a new set of 4D brain imaging data. This new data projects the brain activity from the grey matter onto the white matter, with the connections' strength influencing the outcome. By performing a weighted average of the BOLD signal based on the probability of connection in the white matter and then conducting a standard general linear model, we are assessing whether the involvement of specific white matter tracts is significant. This approach allows for a more nuanced examination of the interplay among various brain regions, moving beyond the traditional focus on their isolated functions in brain processes. Compared with earlier methods that initiated tractography directly from functional activation sites[17], the Functionnectome facilitates a data driven statistical analysis of the implicated white matter pathways.

The present paper focuses on the mechanisms of visual consciousness, acknowledging their association with attentional processes. We employed the Functionnectome in combination with three distinct fMRI paradigms that disentangle attention from conscious perception[18–20]. Specifically, phasic alerting, spatial orienting, or executive attention were manipulated in each paradigm by presenting an alerting tone, a peripheral cue, and a Stroop task, respectively for each process. The target was a near-threshold Gabor

[1]Experimental Psychology Department, and Brain, Mind, and Behavior Research Center (CIMCYC-UGR), University of Granada, Granada, Spain. [2]Groupe d'Imagerie Neurofonctionnelle, Institut des Maladies Neurodégénératives-UMR 5293, CNRS, CEA University of Bordeaux, Bordeaux, France. [3]Sorbonne Université, Institut du Cerveau—Paris Brain Institute—ICM, Inserm, CNRS, APHP, Hôpital de la Pitié-Salpêtrière, Paris, France. [4]Brain Connectivity and Behaviour Laboratory, Sorbonne Université, Paris, France. ✉e-mail: msignes@ugr.es; michel.thiebaut@gmail.com

stimulus individually titrated to be perceived ~50% of the time. Participants had to discriminate the orientation of the Gabor's lines (except in the executive attention task), and to detect and report whether or not they saw the target (see Methods section for a detailed description of the tasks). Through a conjunction analysis of the contrast seen >unseen, we aimed to reveal specific networks that underlie visual consciousness, expanding our understanding of the neural correlates of visual consciousness independently from attention.

We hypothesized that consciousness emerges from the interaction within and between circuits involving key anatomical pathways, including frontoparietal fibers (i.e., the superior longitudinal fasciculus), vertical temporoparietal fibers (i.e., the posterior segment of the arcuate fasciculus), and anterior thalamic radiations. The superior longitudinal fasciculus projections connect the areas showing an increase of blood flow during explicit conscious access and a reduction in metabolism during the loss of consciousness[11], as well as areas usually reported as damaged in patients showing lack of awareness for one side of space (i.e., spatial neglect)[21]. Initial evidence in a stroke patient suggests the importance of temporoparietal fibers in connecting preconscious processing to the frontoparietal systems linked by the superior longitudinal fasciculus[22]. Furthermore, thalamic contributions play a critical role in interregional coupling, facilitating interactions through interregional synchronization[23] (for a review, see ref. 24).

## Results
### fMRI task activations
Figure 1a presents the voxelwise analysis of the fMRI paradigms manipulating visual consciousness, following a classical approach. The conjunction of results revealed the standard cortical network of the global neuronal workspace, encompassing frontal (i.e., left precentral and superior frontal gyrus), parietal (mainly the intraparietal sulcus) and temporal lobes (activation centered onto the posterior portion of the inferior temporal sulcus and the fusiform gyrus).

In contrast, Fig. 1b showcases the same data's Functionnectome analysis, confirming the expected involvement of distinct anatomical pathways. Specifically, the second branch of the superior longitudinal fasciculus, the posterior segment of the arcuate fasciculus, and the anterior thalamic radiations exhibited significant involvement. Furthermore, the splenium of the corpus callosum demonstrated a significant involvement.

### Comparison between classical and Functionnectome analysis approach
Classical activation analysis and Functionnectome analysis represent different models of the brain functioning, one focusing statistically on independent activation of single voxels (i.e., functional segregation) and the other considering the weighted average of activation along white matter connections (i.e., functional interaction). Currently, limited knowledge exists regarding the differences between these two methods for depicting brain functioning.

To explore this question, we created histograms depicting the distribution of z values for each brain voxel using both methods (Fig. 2a). An ANOVA (detailed in the methods section) revealed a strong interaction between the employed techniques and the level of activations in the number of activated voxels ($F_{(1, 154)} = 14.329$, $P = 0.000219$). Interestingly, activations were more prominently represented in the Functionnectome fMRI analyses than in the classical approach, as revealed by a post-hoc independent sample t test ($t_{(77)} = -2.557$, $P = 0.013$). Conversely, de-activations were more frequent in the classical analysis than in the Functionnectome ($t_{(77)} = 2.847$, $P = 0.006$).

## Discussion
Employing state-of-the-art fMRI techniques, we harnessed the power of advanced analyses to elucidate the neural underpinnings of conscious visual perception. By directly comparing the traditional topological approach (classical fMRI) with the cutting-edge hodological perspective (study of pathways via Functionnectome), we suggested a pivotal role of interactions in the conscious access to visual information. The set of results found with the classical fMRI approach was highly congruent with neuroimaging literature showing brain regions involved in eliciting a conscious visual percept (for a recent meta-analysis see ref. 25). However, the Functionnectome approach led us to identify four distinct anatomical circuitry components (see Fig. 2b), each with the potential to give rise to distinct disorders of conscious visual perception when disconnected.

The extensive body of literature consistently underscores the relevance of the frontoparietal cortical network in facilitating conscious perception (for a review see ref. 11. In individuals with intact cognitive function, variations in the strength of frontoparietal connections—the superior longitudinal fasciculus—have been associated with visual perception, leading to either distortions[26] or perceptual enhancements[18,27,28]. When these critical frontoparietal fibers, and especially the second branch of the superior

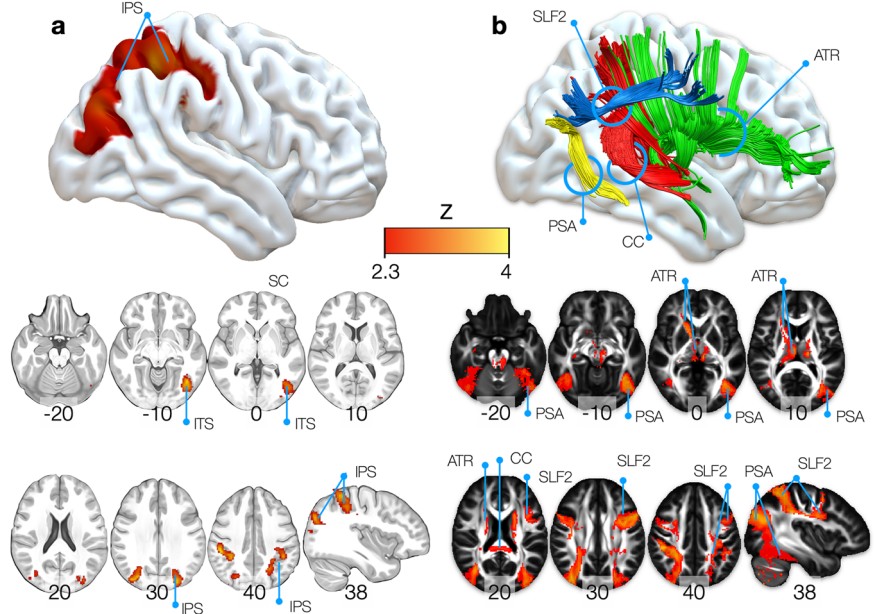

**Fig. 1 | Results maps obtained in the conjunction analysis for the seen > unseen contrast for the classical and Functionnectome approaches.**
**a** Results for the classical analysis approach.
**b** Results for the Functionnectome approach. Maps are corrected at a cluster-defining threshold of Z > 2.3 and a cluster threshold of $P < 0.05$. N = 3 independent experiments, including 18, 18, and 20 participants, respectively. While results from the classical approach replicate brain regions often found in neuroimaging literature, the Functionnectome expands those results to demonstrate the involvement of white matter tracts. Z maps of the results can be found in https://neurovault.org/collections/15553/. ATR anterior thalamic radiation, CC corpus callosum, IPS inferior parietal sulcus, ITS inferior temporal sulcus, PSA posterior segment of the arcuate fasciculus, SLF2 second branch of the superior longitudinal fasciculus.

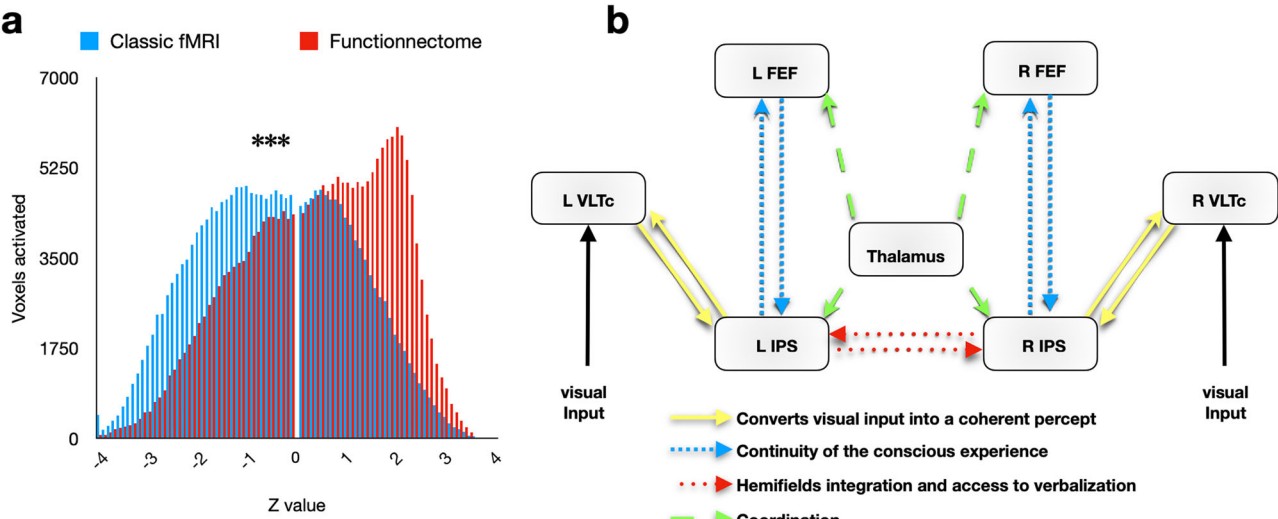

**Fig. 2 | Histogram of z values for the conjunction maps and proposed model of visual conscious access circuit based on the results. a** Distribution of z values for each brain voxel of the conjunction maps calculated using the classical (blue color; N = 216,289 independent voxels) and the Functionnectome (red color; N = 220,157 independent voxels). The dataset to generate the graph can be found at https://osf.io/gb2nh/[52]. **b** Graphical representation of the anatomical connections between relevant nodes of the proposed conscious access circuit. Yellow arrows represent the posterior segment of the arcuate fasciculus; red arrows represent the splenium of the corpus callosum; blue arrows represent the second branch of the superior longitudinal fasciculus; green arrows represent the anterior thalamic radiation. FEF frontal eye field, IPS inferior parietal sulcus, VLTc ventro lateral temporal cortex.

longitudinal fasciculus, become disconnected, as observed in cases of visual neglect, patients demonstrate an absence of awareness regarding events occurring on the contralesional side[29]. Furthermore, temporary neglect induced by electrical perturbation of this white matter pathway during awake brain surgery disrupts the symmetrical processing of visual scenes[21]. Converging evidence on the role of these frontoparietal networks comes from similar studies using magneto-encephalography[30] or intracerebral recordings in drug-resistant epilepsy[31]. Given the existing body of evidence, identifying activation within the networks linked by the second branch of the superior longitudinal fasciculus during conscious visual access aligns seamlessly with our predictions. Moreover, it is essential to note that this network's functionality extends beyond conscious visual perception, as other crucial cognitive processes, such as working memory and representation, colocalize within it[32]. These additional functions play an indispensable role in mentally manipulating concepts as a form of *stream of consciousness*[3], allowing individuals to consciously maintain and report information in a meaningful manner.

Fibers connecting occipital-temporal-parietal regions would transfer visual information from the early visual areas to the visual ventral stream[33]. At that level, the processing of information may be preconscious. Supporting this idea, Dalla Barba and collaborators[22] reported a single patient with left visuospatial neglect symptoms that presented a right temporoparietal lesion that caused a disconnection between the occipitotemporal visual processing stream and the frontoparietal attentional networks. Therefore, a crucial part of the visual-conscious access circuit may be supported by those temporoparietal connections.

In addition to intra-hemispheric fibers, connections between left and right hemispheres may be necessary for transferring the information and building a unified conscious representation[34], particularly through its verbalization[35]. Our analysis revealed a significant involvement of the splenium of the corpus callosum, which are projections from the occipital-parietal and temporal homologous cortices. The splenium of the corpus callosum has been implicated in visual neglect when compared with patients without neglect and in the chronic persistence of the symptoms[36], and in association with symptom severity[37].

The involvement of the medio-dorsal thalamus and its cortical projections through the internal capsule in the visual conscious access circuit is worth considering. The thalamus has been extensively linked to disorders of consciousness, as demonstrated by studies such as Liyana Arachige et al.[38], which highlight its connections with both the voluntary movement motor circuit and the anterior forebrain mesocircuit[39]. In addition to its role in state-based precursors of consciousness, such as states of consciousness and arousal, the thalamus may also play a part in conscious access. Evidence suggests that lesions affecting the laterodorsal portion of the thalamus can lead to spatial neglect in patients with preserved parietal-frontal connections[29]. Moreover, Kronemer et al.[40] demonstrated the existence of a thalamic awareness potential and an increase in BOLD signal within the thalamus following the perception of a visual threshold stimulus, irrespective of whether it involved a report or no-report paradigm. The thalamus's potential critical role in overseeing interregional coupling at the cortical level[23,24] and, accordingly, integrative mechanisms might explain its contribution to visual conscious access.

Our comparative analysis delineates the distinctions between traditional fMRI analyses and the Functionnectome approach in evaluating task-related fMRI signals. This examination reveals that, in contrast to classical analyses that primarily identify isolated activations, the incorporation of brain interactions along white matter pathways provides a significant alternative perspective. This perspective focuses on the exchanges between brain regions during visual-conscious access. Importantly, our analysis detected much less deactivation than activation spreading along white matter pathways. This suggests that deactivation processes may not involve structural connections. Instead, they may be attributed to decreased activation or reduced oxygen consumption[41,42] in specific regions without direct structural links. Therefore, our findings propose that consciousness involves more than localized brain activation or deactivation, highlighting the importance of interactions between brain areas.

In conclusion, our study elucidates that consciousness emerges from a network of interactions predominantly between the brain regions implicated in the global neuronal workspace[11]. Our results suggest that the access to visual consciousness is mediated by a tripartite circuit of connections (Fig. 2b) encompassing the visual ventral stream, which converts visual input into a coherent percept, the frontoparietal networks, which contribute to the continuity of conscious experience, and the interhemispheric connections facilitating left and right hemifields integration and access to verbalization. Within this complex neural network, the thalamus plays a role as a crucial conductor, supervising and monitoring the coordination of these

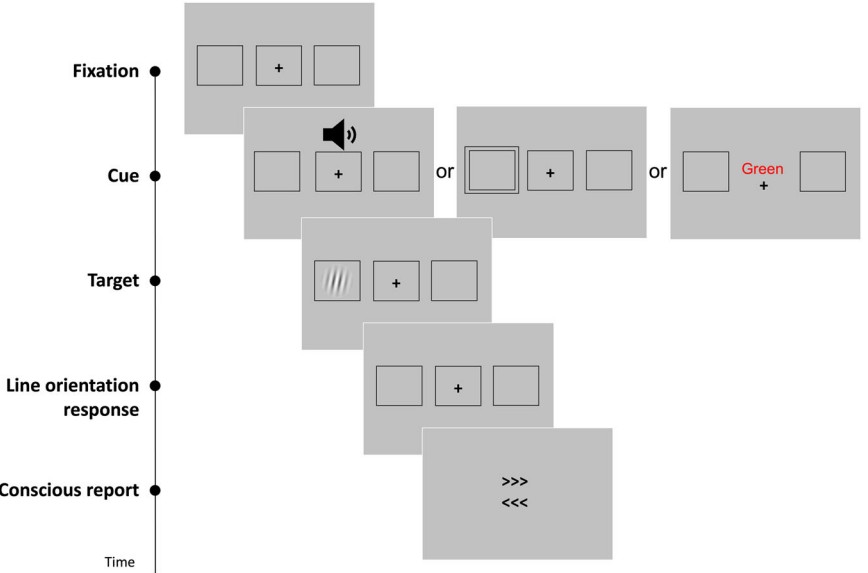

**Fig. 3 | Sequence of events in a given trial of each task.** Attention was manipulated with the presentation of a tone, a peripheral cue, or a Stroop task, for the alerting, orienting and executive attention tasks, respectively. The target was a near-threshold Gabor stimulus, perceived ~50% of the time. Participants had to identify the orientation of the lines of the Gabor, and report whether or not they consciously detected its appearance in one of the markers. Note that for the orienting task, the conscious report was made by indicating if the target was seen or unseen, without reporting target location. For the executive attention task, the Stroop word was presented concurrently to the appearance of the target, and the response to the orientation of the target's lines was not requested. For a detailed figure of each paradigm, please refer to the original publications[18–20].

interconnected processes. The integration of these components sheds light on the fundamental mechanisms that give rise to conscious experience, furthering our understanding of the nature of consciousness.

## Methods

### Neuroimaging data

Three datasets were employed in this work. For the alerting task, we used data from the study conducted by Chica et al.[20], which included 18 participants. Similarly, we employed data from Chica et al.'s study[19] for the orienting task, which also involved 18 participants. Lastly, we utilized data from Martín-Signes et al.'s study[18], which had 20 participants for the executive attention task. Data used in this study was collected with the informed consent of the participants and was approved by the Ethics Committee of the INSERM (France) and the University of Granada (Spain). All ethical regulations relevant to human research participants were followed. The authors of the original studies provided us with the datasets, and we analyzed them with their consent.

### Experimental tasks

To learn more about the experiment, please refer to the original research articles[18–20]. To disentangle the attentional mechanism from conscious perception, we combined data from three visual tasks with different types of attentional manipulations (phasic alerting, spatial orienting, and executive attention). The three tasks involved identifying a near-threshold stimulus that could appear in one of two lateral boxes on the screen. In some trials, the stimulus was not presented. The contrast of the stimulus was adjusted for each participant before the task to ensure that they perceived it about half of the time. For a schematic representation of the sequence of events in a given trial of each task, please see Fig. 3.

In the alerting task, a white noise cue was played on half of the trials. Participants had to make two consecutive responses: first, they identified the orientation of the lines in the stimulus, and second, they reported whether or not they consciously detected it by indicating its location (right or left box) or indicating that the target was not seen. The experiment consisted of two sessions with five functional scans each, lasting 12 min each, for 920 trials.

During the orienting task, a square would appear around one of the boxes on the side of the screen, indicating where the target would likely be located on 67% of the trials. This cue was displayed for 300 ms. Similar to the alerting task, participants were required to give two consecutive responses: identifying the orientation of the Gabor's lines and reporting if they consciously detected the target's appearance. The experiment had one session of 5 functional scans, each 7 min long, with 280 trials.

During the executive attention task, participants were presented with a Stroop task. This task involved Spanish words for blue, green, and yellow colors displayed in blue, green, or yellow. If the word meaning and the color matched, it was called a congruent trial. If they were different, it was called an incongruent trial (which happened 20% of the time). Participants were asked to do two consecutive tasks: first, they had to discriminate the word's color, and second, they had to report if they consciously detected the target's appearance and its location. The experiment consisted of 2 sessions with five functional scans of 8 min each, resulting in 600 trials.

In the three experiments, participants underwent a titration procedure that ensured that ~50% would be "seen trials" (i.e., participants reported that they consciously perceived the target and correctly located it), while the other ~50% were "unseen" (i.e., participants indicated that they did not perceive the target). Indeed, when the target was consciously detected, the discrimination response (i.e., identifying the orientation of the Gabor's lines) was above chance, while it was at chance when it was not detected. The awareness main effect was significant for both the alerting and orienting tasks (alerting task: $F(1, 18) = 281.46$, $P < 0.001$, discrimination accuracy for seen reports $M = 76\%$, and unseen reports $M = 46\%$; orienting task: $F(1,17) = 195.69$, $P < 0.001$, discrimination accuracy for seen reports $M = 93\%$, and unseen reports $M = 47\%$). Note that the executive attention task did not include a discrimination response. In addition, the percentage of false alarms (i.e., reporting to detect the target when it was not presented) was kept very low (alerting task: 13% of the target-absent trials, SD = 26.20; orienting task: 6.91% of the target-absent trials, SD = 9.19; executive task: 4.9% of the target-absent trials, SD = 6.86).

During the study, participants were shown the task on a screen at the back of the scanner via a mirror mounted on the head coil. They

responded by pressing buttons on an MRI-compatible fiber optic box. The order of the trial types and the jitter fixation were determined by an optimal sequencing program called Optseq2. This program is designed to maximize the efficiency of recovering the blood-oxygen-level dependent (BOLD) response, which is a reliable indicator of brain activity[43].

## Acquisition parameters
The acquisition parameters used for the whole-brain fMRI study can be found in the original publications by Chica et al.[19,20] and Martín-Signes et al.[18]. The studies were conducted on two different Tesla Siemens TRIO MRI scanners using a whole-head coil. Functional images were acquired using a gradient-echo echo-planar pulse sequence. For the alerting task, they used 372 volumes acquired per run with the time-to-repetition (TR) set at 2000 ms, the time-to-echo (TE) at 25 ms, using 39 axial 3-mm cubic slices with no inter-slice gap, a flip angle set at 75°, and the field of view (FoV) of 220 mm. For the orienting task, they used 220 volumes per run with a TR set at 2000 ms, a TE at 25 ms, 34 axial $2.5 \times 2.5 \times 3$-mm slices with no inter-slice gap, a flip angle set at 75°, and a FoV of 220 mm. For the executive attention task, they used 245 volumes per run with a TR set at 2000 ms, a TE at 25 ms, 35 axial 3.4-mm cubic slices with no inter-slice gap, a flip angle set at 75°, and a FOV of 220 mm. In addition, high-resolution T1-weighted anatomical images were systematically collected.

## fMRI preprocessing
We employed FEAT (FSL, FMRIB's Software Library, Woolrich et al.[44]) to perform preprocessing routines and analyses. In short, those included brain extraction[45], slice timing correction, and realignment through rigid-body transformation for motion correction using MCFLIRT[46]. Motion plots were visually inspected to discard those runs with excessive motion (i.e., relative motion > half of voxel size and/or absolute motion > voxel size). Structural and functional volumes of each participant were coregistered using the Boundary-Based Registration function. Next, we registered the structural volume to a standard image, and a similar transformation was applied to the functional volume using a non-linear registration with 12 degrees of freedom. During normalization to the MNI152 stereotaxic space, volumes were upsampled to 2 mm isotropic voxels. A 128 s high-pass filter was used to eliminate contamination from slow signals drift. Outlier scans corrupted by large motion were detected using the tool fsl_motion_outliers and regressed out. The number of outlier scans never exceeded 20% of the total scans in a run. For the classical approach, the signal from neighboring voxels was combined applying a 5-mm full width at half maximum (FWHM) Gaussian smoothing[47]. In contrast, the Functionnectome approach required no spatial smoothing as it already combines the signal from distant yet structurally linked voxels[15].

## Functionnectome approach
We used the Functionnectome (openly available at http://www.bcblab.com) to process preexisting data[18–20]. The Functionnectome projects the signal from each voxel of the fMRI volume (with four dimensions; 4D) to the white matter according to their structural relationships. These structural relationships were based on a probability map that depicts the structural connectivity between a particular voxel and the rest of the brain. This map is derived from a normative high-resolution tractography resource acquired at 7 T in 100 subjects[48] derived from the processed version[49] of the human connectome project (raw data available at www.humanconnectome.org and processed tractographies at https://osf.io/5zqwg/). The Functionnectome utilizes the structural connectivity data provided by white matter priors to calculate a weighted average of the BOLD time series originating from the grey matter voxels connected to a specific white matter voxel. As a result, the Functionnectome generates a new 4D volume that projects the fMRI signal from grey matter voxels to white matter, weighted by the connections' probability. This 4D functional volume, with functional time series on the white matter voxels, can be statistically analyzed similarly to a classical fMRI volume[15].

## Statistics and reproducibility
For both the classical and the Functionnectome approaches, we used the general linear model and convolved task regressors with the FSL double-gamma function to analyze each run. For the three tasks, fMRI trials were categorized as either "seen" or "unseen" based on participants' responses.

We included six head motion parameters and outlier scans as regressors of no interest to avoid motion artifacts in all first-level analyses for the three tasks. We calculated intra-subject brain activations for the contrast seen > unseen using fixed effects and conducted higher-level mixed-effects using FLAME 1[50]. In order to minimize the contamination of the results by the attentional mechanism, a conjunction analysis using *easythresh_conj* tool from FSL[51] was applied to the three seen > unseen statistical maps derived from the three experimental paradigms (phasic alerting, spatial orienting, and executive attention). Note that these conjunction maps were calculated independently using the contrast maps derived from each analysis approach. Correction for multiple comparisons was conducted at a cluster-defining threshold of $Z > 2.3$ and a cluster threshold of $P < 0.05$ to render Z-statistic BOLD images. Results were interpreted and labeled by an expert neuroanatomist (M.T.d.S.).

In order to compare fMRI classical and Functionnectome outcomes, we constructed a histogram showcasing the count of activated voxels for each z value ranging from z = -4 to z = +4, at increments of 0.1. Following this, to evaluate any statistical difference between the methods, we conducted an ANOVA test with voxel count as the independent variable and the discretized z value as the dependent variable.

## Reporting summary
Further information on research design is available in the Nature Portfolio Reporting Summary linked to this article.

## Data availability
All fMRI results maps and dataset of the Z-values histogram are publicly available via OSF (https://osf.io/gb2nh/[52]). Z maps of the conjunction results are additionally available via Neurovault for a convenient visualization (https://neurovault.org/collections/15553/). The conditions of the ethics approvals of the datasets do not permit public archiving or sharing of anonymized raw study data.

## Code availability
Analyses were carried out using open software and toolboxes (see methods section). The Functionnectome is an open-source software available at http://www.bcblab.com. All fMRI preprocessing routines and analyses were performed with FEAT, part of FSL (FMRIB's Software Library, http://www.fmrib.ox.ac.uk/fsl). Conjunction analysis was performed with the easy-thresh_conj FSL's tool (https://warwick.ac.uk/fac/sci/statistics/staff/academic-research/nichols/scripts/fsl/easythresh_conj.sh).

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

## Acknowledgements

M.M.S. is supported by a Margarita Salas fellowship by the Spanish Ministry of Universities and the European Next Generation funding, and by a contract

for Young Researchers (PAIDI 2020) by the Ministry of Economy, Knowledge, Enterprise, and Universities of Andalusia. A.B.C. is supported by research project PID2020-119033 GB-I00 funded by MCIN/AEI/10.13039/501100011033, and by "ERDF A way of making Europe", by the European Union; and by FEDER/Junta de Andalucía-Consejería de Economía y Conocimiento/ project A.SEJ.090. UGR18. P.B. is supported by the *Agence Nationale de la Recherche* through ANR-16-CE37-0005 and ANR-10-IAIHU-06, and by the *Fondation pour la Recherche sur les AVC* through FR-AVC-017. M.T.d.S is supported by HORIZON- INFRA-2022 SERV (Grant No. 101147319) "EBRAINS 2.0: A Research Infrastructure to Advance Neuroscience and Brain Health", by the European Union's Horizon 2020 research and innovation programme under the European Research Council (ERC) Consolidator grant agreement No. 818521 (DISCONNECTOME), the University of Bordeaux's IdEx 'Investments for the Future' program RRI 'IMPACT', and the IHU 'Precision & Global Vascular Brain Health Institute – VBHI' funded by the France 2030 initiative (ANR-23-IAHU-0001).

## Author contributions

M.M.S. and M.T.d.S.: conceptualization, formal analysis, writing—original draft, writing—review and editing. A.B.C. and P.B.: writing—review and editing.

## Competing interests

The authors declare no competing interests. M.T.d.S. is an Editorial Board Member for *Communications Biology* but was not involved in the editorial review of, nor the decision to publish this article.
