## [Peer Review file · Communications Biology]

Streams of conscious visual experience

Corresponding Author: Dr Mar Martín-Signes

Version 0:

Reviewer comments:

Reviewer #1

(Remarks to the Author)

This is a valuable contribution by Martín-Signes and collaborators. The study is well motivated and the results are very interesting. My only recommendation is that the paper is extremely short and could use further elaboration.

The paper is largely centered on the novel methodology the authors call "functionnectome". Given that most readers will not be familiar with the approach, I would encourage devoting a paragraph or two in the Introduction to present the approach further. I would also strongly encourage the authors to explain the methodology in more detail in the Methods section. As it currently stands, it just provides an extremely short summary.

Relatedly, the authors state that the results demonstrate the role of integration. I am not familiar with the methodology the authors used but why does white-matter tract involvement demonstrate integration? I think I know what the authors have in mind, but the involvement of fiber tracts could potentially indicate signal "communication", "distribution", and so on. In what way does it imply "integration" which I take to mean something closer to having multiple sources of information combined in ways that go beyond just signal transmission.

Given the challenges of establishing visual awareness, what was the criterion adopted? Better than chance performance on the yes/no question? Was awareness also evaluated with signal detection theory? If not, isn't response bias a concern? Because the paper is about "streams of conscious" experience, further discussion would be warranted.

Additional questions.

- What procedure for cluster significance was utilized in the maps of Figure 1?
- Line 93: do the authors mean strong "intersection"?

Reviewer #2

(Remarks to the Author)

The manuscript called "Streams of conscious visual experience" by Martín-Signes et al. presents a reanalysis of databases studying conscious perception using the novel method called functionnectome. They find that the new method replicates results found by classical methods and possibly adds information that classical methods, based on the analysis of individual voxels, miss. I find that the paper is valuable and interesting, and the proposed new method (already published by some of the authors of the present work) is novel and has great potential. I have some comments to improve the current work.

1- The text is written in a very unclear manner for my taste, with almost poetic phrases like "Consciousness encompasses the essence of sentient beings," "holistic understanding," "symphony of neural connections that culminate in awareness," and others of the like. While I agree that the study of consciousness often has a metaphorical and approximate tone (since we do not know the mechanism by which consciousness arises), I think it is excessive in this work, and the work does not benefit from so much metaphor. I would describe the findings more succinctly, also not relating them so much to the IIG and GNW theories, which are not questioned or challenged, and the results do not lean towards one or the other.

- In the text and figures, I would emphasize more on the aspects revealed by the new method that the traditional method cannot account for.

- In the histogram of Figure 2, it can be seen that the methods differ in the distributions of z values. What does this mean? Why would it be good to have more or less? The figure shows that the methods are different, which is expected because the methods are radically different. Figure 2b is a scheme, without anything quantifiable. There is likely very interesting information there about the contrast 'seen vs. not seen' and the networks involved, but it is not clear.

Reviewer #3

(Remarks to the Author)

This paper reports on a data analysis of existing fMRI data to elucidate what major white matter tracts might be involved in the integrative mechanisms presumed to underly visual consciousness. Through a conjunction analysis, the study pooled data from three studies that probed different aspects of visual conscious processing (controlling for attentional mechanisms) to map which white matter voxels are involved across these studies. This was done using a novel approach called Functionnectome analysis, wherein the BOLD timeseries data in the brain's gray matter were projected onto the related white matter. Specifically, as defined in the original methods paper: the method computes "a weighted average, on a given voxel, of the BOLD signal from the voxels sharing a structural link (given by the anatomical priors) to this voxel. [...] In other words, if we focus on a single functionnectome voxel v , the value of this voxel is equal to the sum of the BOLD signal from every voxel in the brain, weighted by the probability of their connection to v (which is 0 if they are not part of the involved circuit), and divided by the sum of all those probabilities". Based on this analysis, the authors found several white matter tracts that are commonly involved in the three datasets: SLF within the fronto-parietal fibers, temporal-parietal fibers and the splenium of the corpus callosum. In addition they found Functionnectome activations in the thalamus. While the findings of the paper are interesting and seem plausible, there are several issues with this manuscript that I believe should be addressed before publication:

1. The main text of the manuscript is too concise. To understand what's going on, and to evaluate the methods and interpretation of the results, the reader currently needs to delve into the three articles that describe the experiments and the Functionnectome analysis is also described without much detail. I believe the manuscript would improve a lot if the three datasets and methods are summarized before presenting the results (detailed methods can still be presented in the methods section following results and discussion). When discussing the datasets, it is also important to explain what these data tell us about conscious processing.
2. The results are over-interpreted. For instance, the authors claim to have "demonstrated the pivotal role of integration in the conscious access to visual information", and many more examples of such over-interpretation can be found in the discussion. The discussion should be revised so that it stays more closely to what the findings really are: additional evidence that the hypothesized white matter tracts are indeed important for conscious visual processing. Please also explain how the results could have been any different: couldn't we have known this in advance given the pattern of gray matter activations? It is not obvious how the analysis could, for instance, arbitrate between competing hypotheses of say one connecting white matter tract being involved but not another. After all, the Functionnectome does not measure involvement of white matter tracts but extrapolates from gray matter data, and the set of tracts that are highlighted are uniquely defined by the WM prior and GM data, which limits what types of questions can be answered with the method.
3. It is not clear what the results presented in figure 2 and associated text are meant to convey. These results are introduced by saying that limited knowledge exists regarding which method (classical vs Functionnectome) better depicts brain functioning, but it is not clear by what standard. It appears that the authors define "better" in terms of the distribution of z-values, but it is not clear to me why one distribution would be better than another. The authors seem to conclude superior explanatory power by the Functionnectome analysis based on an increased number of positive z-values, but that is not valid because a better (i.e. true) model could very well have fewer activated voxels. Moreover, the number of positive z-values is expected to increase if the analysis selectively projects time-series into the white matter from a region of interest (mask M as defined in the original methods paper) with high z-values for the conjunction contrast (if this is not the case, please explain this better in the methods).

Minor: Please insert explicit Results caption.

Author Rebuttal letter:

Reviewer #1

This is a valuable contribution by Martin-Signes and collaborators. The study is well motivated and the results are very interesting. My only recommendation is that the paper is extremely short and could use further elaboration.

We really appreciate the positive comments on our study. We also thank the reviewer for the time invested in reviewing the manuscript and for the further comments. Below, we respond one-by-one to Reviewer #1's comments, which have also been addressed in the revised version of the manuscript (highlighted in blue text).

Comment 1: The paper is largely centered on the novel methodology the authors call "functionnectome". Given that most readers will not be familiar with the approach, I would encourage devoting a paragraph or two in the Introduction to present the approach further. I would also strongly encourage the authors to explain the methodology in more detail in the Methods section. As it currently stands, it just provides an extremely short summary.

Thank you for the suggestion. We have incorporated a new paragraph into the Introduction to provide readers with an overview of the Functionnectome approach. Additionally, we have expanded this explanation in the Methods section.

Added to the introduction (pages 2-3, lines 51-70): âThe introduction of the Functionnectome represents a new methodology that integrates structural connectivity data within functional analysis. The Functionnectome takes the activity signals (BOLD time series) from the grey matter and combines them based on how these grey areas are connected to white matter areas. The strength of each connection influences the final signal. As a result, it creates a new type of 4D brain imaging data. This new data projects the brain activity from the grey matter onto the white matter, with the connections' strength influencing the outcome. This approach allows for a more nuanced examination of the interplay among various brain regions, moving beyond the traditional focus on their isolated functions in brain processes. Compared with earlier methods that initiated tractography directly from functional activation sites (Javad et al., 2014), the Functionnectome facilitates a data driven statistical analysis of the implicated white matter pathways. Moreover, it addresses a critical limitation of conventional functional fMRI analyses, which typically employ spatial filtering prior to statistical evaluation. Such traditional practices do not consider the structural connections between voxels and may inadvertently merge signals from functionally disparate regions. By eliminating the necessity for spatial filtering, the Functionnectome offers a pathway to conduct statistical analyses with a heightened degree of precision and sensitivity, thereby holding the potential to unveil the white matter circuits that underpin brain interactions.â

Added to the methods (pages 15-16, lines 356-372): âWe used the Functionnectome (openly available at <http://www.bcblab.com>) to process preexisting data (Chica et al., 2013, 2016; MartÃn-Signes, Paz-Alonso, et al., 2019). The Functionnectome projects the signal from each voxel of the fMRI volume (with four dimensions; 4D) to the white matter according to their structural relationships. These structural relationships were based on a probability map that depicts the structural connectivity between a particular voxel and the rest of the brain. This map is derived from a normative high-resolution tractography resource acquired at 7T in 100 subjects (Vu et al., 2015) derived from the processed version (Thiebaut de Schotten et al., 2020) of the human connectome project (raw data available at www.humanconnectome.org and processed tractographies at <https://osf.io/5zqwg/>). The Functionnectome utilizes the structural connectivity data provided by white matter priors to calculate a weighted average of the BOLD time series originating from the grey matter voxels connected to a specific white matter voxel. As a result, the Functionnectome generates a new 4D volume that projects the fMRI signal from grey matter voxels to white matter, weighted by the connectionsâ probability. This produces a new 4D functional volume, with functional time series on the white matter voxels that can be statistically analyzed similarly to a classical fMRI volume (Nozais et al., 2021).â

Comment 2: Relatedly, the authors state that the results demonstrate the role of integration. I am not familiar with the methodology the authors used but why does white-matter tract involvement demonstrate integration? I think I know what the authors have in mind, but the involvement of fiber tracts could potentially indicate signal âcommunicationâ, âdistributionâ, and so on. In what way does it imply âintegrationâ which I take to mean something closer to having multiple sources of information combined in ways that go beyond just signal transmission.

We appreciate the opportunity to discuss further our reasoning behind using the term "integration" and our decision to amend it to "interaction." In our original manuscript, we employed "integration" to describe the complex processes within the brain that go beyond mere signal transmission, aiming to highlight the intricate ways in which different sources of information are combined and processed. This choice was grounded in our belief that the interactions facilitated by white matter tracts represent integrative mechanisms, allowing for the synthesis of information across various cortical regions. Specifically, we referred to how the Functionnectome approach integrates data, performing a weighted average of values from connected cortical areas based on connection probabilities, thereby illustrating that the brain's output is more than just the sum of its parts. However, we acknowledge the point that the term "integration" might imply a level of combinatorial processing that cannot be distinctly separated from simpler forms of communication, especially given the limitations in temporal resolution of fMRI data. This insight made us reconsider our terminology, opting for "interaction" to describe these processes. This term, we believe, more accurately captures the nature of the white matter tract's role in facilitating connections between brain regions without overstating the complexity of these interactions. Consequently, we have revised our manuscript to replace "integration" with "interaction" throughout to more precisely reflect the current understanding and the evidence provided by our data.

Comment 3: Given the challenges of establishing visual awareness, what was the criterion adopted? Better than chance performance on the yes/no question? Was awareness also evaluated with signal detection theory? If not, isnât response bias a concern? Because the paper is about âstreams of consciousâ experience, further discussion would be warranted.

We appreciate the reviewer for bringing up this point. In the three experiments, Gabor stimuli were individually titrated for each participant to achieve a conscious perception of approximately 50% of the stimuli. This titration procedure was based on the detection response, where participants had to report whether they consciously detected the appearance of the target. Indeed, when the target was consciously detected and reported, the discrimination response (i.e., identifying the orientation of the Gabor lines) was above chance while it was at chance when it was not detected (Awareness main effect for the alerting task: $F(1, 18) = 281.46$, $p < 0.001$, discrimination accuracy for seen reports $M = 76\%$, and unseen reports $M = 46\%$; and the orienting task: $F(1, 17) = 195.69$, $p < 0.001$, discrimination accuracy for seen reports $M = 93\%$, and unseen reports $M = 47\%$. Note that the executive attention task did not include a discrimination response). Although perceptual sensitivity and response bias could have been calculated overall or for each attentional condition, this was not done in the alerting and orienting experiments. In the executive attention experiment, both indices were calculated and reported but were not modulated by executive attention. Indeed, while other experimental approaches had manipulated awareness by including stimuli with different perceptual sensitivity and response bias (e.g., (Fernández et al., 2023; Jigo & Carrasco, 2020), in these experiments awareness was manipulated and measured in a trial by trial manner. In addition, efforts were made to minimize the number of false alarms. Participants were instructed to report stimuli only when they were confident about their perception, thus reducing the risk of adopting a liberal criterion and resulting in a high number of false alarms. As a result, the percentage of false alarms was very low (alerting task: 13% of the target-absent trials, $SD = 26.20$; orienting task: 6.91% of the target-absent trials, $SD = 9.19$; executive task: 4.9% of the target-absent trials, $SD = 6.86$). We have now included this information in the Method section (pages 12-13, lines 291-299; and footnotes 1 and 2, pages 12 and 13).

Additional question 1: What procedure for cluster significance was utilized in the maps of Figure 1?

Higher-level mixed-effects analyses were performed using FLAME 1 (FMRIB's Local Analysis of Mixed Effects). Concretely, FLAME 1 is a sophisticated method using Bayesian modeling and estimation (Woolrich et al., 2004). Correction for multiple comparisons was conducted at a cluster-defining threshold (CDT) of $Z > 2.3$ and a cluster threshold of $p < 0.05$. This approach is recommended by FSL developers and has been used in e.g. (Haweel et al., 2021; Miller et al., 2019; Sanz-Arigita et al., 2021; Zuber et al., 2020). The methodology has been clarified in Figure 1's caption and in the Methods section.

The revised sentence in Figure 1's caption (page 5, lines 116-121) now reads: "Results maps obtained in the conjunction analysis for the seen > unseen contrast with the classical (a) and the Functionnectome (b) approaches at a cluster-defining threshold of $Z > 2.3$ and a cluster threshold of $p < 0.05$."

Added in the Methods (page 16, lines 379-389): "We calculated intra-subject brain activations for the contrast seen > unseen using fixed effects and conducted higher-level mixed-effects using FLAME 1 (Woolrich et al., 2004)." and "Correction for multiple comparisons was conducted at a cluster-defining threshold of $Z > 2.3$ and a cluster threshold of $p < 0.05$ to render Z-statistic BOLD images."

Additional question 2: do the authors mean strong "intersection" (Line 93)?

Thank you for highlighting the ambiguity surrounding our use of the term "interaction" in line 93. Upon review, we realize that our initial wording may have inadvertently led to confusion regarding its meaning in the context of our analysis. To clarify, when we referred to "interaction" in this context, we intended to describe a statistical interaction as revealed through ANOVA analysis, rather than an "intersection" of methods or data sets. This interaction pertains to the significant effect that the employed fMRI analysis approaches have on the observed levels of activation across different voxels. We have now expanded the Methods section to provide a detailed description of the ANOVA analysis process we followed.

The revised sentence in the Results section (page 6, lines 134-137) now reads: "An ANOVA analysis (detailed in the methods section) revealed a strong interaction between the employed techniques and the level of activations in the number of activated voxels ($F(1, 154) = 14.329$, $p = 0.000219$).

In the methods (page 16, lines 391-396) we specified the following: "In order to compare fMRI classical and Functionnectome outcomes, we constructed a histogram showcasing the count of activated voxels for each z value ranging from $z = -4$ to $z = +4$, at increments of 0.1. Following this, to evaluate any statistical difference between the methods, we conducted an ANOVA test with voxel count as the independent variable and the discretized z value as the dependent variable."

References

- Fernández, A., Hanning, N. M., Carrasco, M., Battelli, L., Chica, A. B., & Mattingley, J. B. (2023). Transcranial magnetic stimulation to frontal but not occipital cortex disrupts endogenous attention. *PNAS*, 120. <https://doi.org/10.1073/pnas.2219635120>
- Haweel, R., Shalaby, A., Mahmoud, A., Seada, N., Ghoniemy, S., Ghazal, M., Casanova, M. F., Barnes, G. N., & El-Baz, A. (2021). A robust DWT-CNN-based CAD system for early diagnosis of autism using task-based fMRI. *Medical Physics*, 48(5), 2315-2326. <https://doi.org/10.1002/MP.14692>
- Jigo, M., & Carrasco, M. (2020). Differential impact of exogenous and endogenous attention on the contrast sensitivity function across eccentricity. *Journal of Vision*, 20(6), 1-25. <https://doi.org/10.1167/JOV.20.6.11>
- Miller, A. B., Prinstein, M. J., Munier, E., Machlin, L. S., & Sheridan, M. A. (2019). Emotion Reactivity and Regulation in Adolescent Girls Following an Interpersonal Rejection. *Journal of Cognitive Neuroscience*, 31(2), 249-261. https://doi.org/10.1162/JOCN_A_01351
- Sanz-Arigita, E., Daviaux, Y., Joliot, M., Dilharreguy, B., Micoulaud-Franchi, J. A., Bioulac, S., Taillard, J., Philip, P., & Altena, E. (2021). Brain reactivity to humorous films is affected by insomnia. *Sleep*, 44(9). <https://doi.org/10.1093/SLEEP/ZSAB081>
- Woolrich, M. W., Behrens, T. E. J., Beckmann, C. F., Jenkinson, M., & Smith, S. M. (2004). Multilevel linear modelling for FMRI group analysis using Bayesian inference. *NeuroImage*, 21(4), 1732-1747. <https://doi.org/10.1016/j.neuroimage.2003.12.023>
- Zuber, P., Tsagkas, C., Papadopoulou, A., Gaetano, L., Huerbin, M., Geiter, E., Altermatt, A., Parmar, K., Ettlin, T., Schuster-Amft, C., Suica, Z., Alrasheed, H., Wuerfel, J., Kesselring, J., Kappos, L., Sprenger, T., & Magon, S. (2020). Efficacy of inpatient personalized multidisciplinary rehabilitation in multiple sclerosis: behavioural and functional imaging results. *Journal of Neurology*, 267(6), 1744-1753. <https://doi.org/10.1007/S00415-020-09768-6>

Reviewer #2

The manuscript called "Streams of conscious visual experience" by Martín-Signes et al. presents a reanalysis of databases studying conscious perception using the novel method called functionnectome. They find that the new method replicates results found by classical methods and possibly adds information that classical methods, based on the analysis of individual voxels, miss. I find that the paper is valuable and interesting, and the proposed new method (already published by some of the authors of the present work) is novel and has great potential. I have some comments to improve the current work.

We thank the reviewer for the time invested in reviewing the manuscript, the positive feedback on our work, and for the further comments. Below, we respond one-by-one to each of Reviewer #2's comments, which have also been addressed in the revised version of the manuscript (highlighted in blue text).

Comment 1: The text is written in a very unclear manner for my taste, with almost poetic phrases like "Consciousness encompasses the essence of sentient beings," "holistic understanding," "symphony of neural connections that culminate in awareness," and others of the like. While I agree that the study of consciousness often has a metaphorical and approximate tone (since we do not know the mechanism by which consciousness arises), I think it is excessive in this work, and the work does not benefit from so much metaphor. I would describe the findings more succinctly, also not relating them so much to the IIG and GNW theories, which are not questioned or challenged, and the results do not lean towards one or the other.

Thank you for your feedback regarding the clarity and style of the text. While our intention was to reflect the intricate nature of consciousness, we recognize that it may have obscured the clarity of the message on this occasion. We have revisited the presentation of ideas in the work to ensure that the essence of the discussion remains clear without compromising depth. We also described the results more succinctly, not relating them to the IIG and GNW theories as per your request.

Comment 2: In the text and figures, I would emphasize more on the aspects revealed by the new method that the traditional method cannot account for.

Thank you for the constructive suggestion. After reviewing our work, as suggested, we acknowledge that this idea was not clearly reflected in the text. We have added some paragraphs that express and emphasize the additional information that the Functionnectome approach, compared to classical fMRI analysis, can provide.

Figure 1's caption now reads (page 5, lines 116-121): "Results maps obtained in the conjunction analysis for the seen > unseen contrast with the classical (a) and the Functionnectome (b) approaches at a cluster-defining threshold of $Z > 2.3$ and a cluster threshold of $p < 0.05$. While results from the classical approach replicate brain regions often found in neuroimaging literature, the Functionnectome expands those results to demonstrate the involvement of white matter tracts."

Added to the Discussion (pages 7-8, lines 154-164): "Employing state-of-the-art fMRI techniques, we harnessed the power of advanced analyses to elucidate the neural underpinnings of conscious visual perception. By directly comparing the traditional topological approach (classical fMRI) with the cutting-edge hodological perspective (study of pathways via Functionnectome), we suggested a pivotal role of interactions in the conscious access to visual information. The set of results found with the classical fMRI approach was highly congruent with neuroimaging literature showing brain regions involved in eliciting a conscious visual percept (for a recent meta-analysis see (MacLean et al., 2023). However, the Functionnectome approach led us to identify four distinct anatomical circuitry components (see Figure 2b), each with the potential to give rise to distinct disorders of conscious visual perception when disconnected."

Comment 3: In the histogram of Figure 2, it can be seen that the methods differ in the distributions of z values. What does this mean? Why would it be good to have more or less? The figure shows that the methods are different, which is expected because the methods are radically different. Figure 2b is a scheme, without anything quantifiable. There is likely very interesting information there about the contrast 'seen vs. not seen' and the networks involved, but it is not clear.

We apologize for the lack of clarity. We now discuss in detail the differences between methods highlighted in Figure 2a as follows (page 10, lines 220-232): "Our comparative analysis delineates the distinctions between traditional fMRI analyses and the Functionnectome approach in evaluating task-related fMRI signals. This examination reveals that, in contrast to classical analyses that primarily identify isolated activations, the incorporation of brain interactions along white matter pathways provides a significant alternative perspective. This perspective focuses on the exchanges between brain regions during visual conscious access. Importantly, our analysis detected much less deactivation than activation spreading along white matter pathways. This suggests that deactivation processes may not involve structural connections. Instead, they may be attributed to decreased activation or reduced oxygen consumption (Logothetis, 2008; Logothetis et al., 2001) in specific regions without direct structural links. Therefore, our findings propose that consciousness involves more than localized brain activation or deactivation, highlighting the importance of interactions between brain areas."

Regarding Figure 2b, our purpose was to summarize our results and interpretation. We now made this clearer in the figure (please, see the updated version of Figure 2b, page 7) and in the text in the discussion as follows (page 10, lines 235-242): "Our results suggest that the access to visual consciousness is mediated by a tripartite circuit of connections (Figure 2b) encompassing the visual ventral stream, which converts visual input into a coherent percept, the fronto-parietal networks, which contribute to the continuity of conscious experience, and the interhemispheric connections facilitating left and right hemifields integration and access to verbalization. Within this complex neural network, the thalamus plays a role as a crucial conductor, supervising and monitoring the coordination of these interconnected processes."

Reviewer #3

This paper reports on a data analysis of existing fMRI data to elucidate what major white matter tracts might be involved in the integrative mechanisms presumed to underly visual consciousness. Through a conjunction analysis, the study pooled data from three studies that probed different aspects of visual conscious processing (controlling for attentional mechanisms) to map which white matter voxels are involved across these studies. This was done using a novel approach called Functionnectome analysis, wherein the BOLD timeseries data in the brain's gray matter were projected onto the related white matter. Specifically, as defined in the original methods paper: the method computes "a weighted average, on a given voxel, of the BOLD signal from the voxels sharing a structural link (given by the anatomical priors) to this voxel. [â]" In other words, if we focus on a single functionnectome voxel v , the

value of this voxel is equal to the sum of the BOLD signal from every voxel in the brain, weighted by the probability of their connection to v (which is 0 if they are not part of the involved circuit), and divided by the sum of all those probabilities. Based on this analysis, the authors found several white matter tracts that are commonly involved in the three datasets: SLF within the fronto-parietal fibers, temporal-parietal fibers and the splenium of the corpus callosum. In addition they found Functionnectome activations in the thalamus. While the findings of the paper are interesting and seem plausible, there are several issues with this manuscript that I believe should be addressed before publication.

Thank you for the positive remarks on our work. We also thank the reviewer for the time invested in reviewing the manuscript and for the further comments. Below, we respond one-by-one to Reviewer #3's comments, which have also been addressed in the revised version of the manuscript (highlighted in blue text).

Comment 1: The main text of the manuscript is too concise. To understand what's going on, and to evaluate the methods and interpretation of the results, the reader currently needs to delve into the three articles that describe the experiments and the Functionnectome analysis is also described without much detail. I believe the manuscript would improve a lot if the three datasets and methods are summarized before presenting the results (detailed methods can still be presented in the methods section following results and discussion). When discussing the datasets, it is also important to explain what these data tell us about conscious processing.

We agree with the reviewer that the main text of the manuscript lacks important details concerning the experiments and the Functionnectome method, which may hinder reader comprehension. We have now included a paragraph in the introduction to provide an overview of the experiments' methods and the Functionnectome approach. Additionally, we have added more details to the Method section regarding how visual conscious processing was measured in the tasks, and a figure (please, see Figure 3, page 13) that illustrates the employed paradigms.

Added to the introduction (pages 2-3, lines 51-70): The introduction of the Functionnectome represents a new methodology that integrates structural connectivity data within functional analysis. The Functionnectome takes the activity signals (BOLD time series) from the grey matter and combines them based on how these grey areas are connected to white matter areas. The strength of each connection influences the final signal. As a result, it creates a new type of 4D brain imaging data. This new data projects the brain activity from the grey matter onto the white matter, with the connections' strength influencing the outcome. This approach allows for a more nuanced examination of the interplay among various brain regions, moving beyond the traditional focus on their isolated functions in brain processes. Compared with earlier methods that initiated tractography directly from functional activation sites (Javad et al., 2014), the Functionnectome facilitates a data driven statistical analysis of the implicated white matter pathways. Moreover, it addresses a critical limitation of conventional functional fMRI analyses, which typically employ spatial filtering prior to statistical evaluation. Such traditional practices do not consider the structural connections between voxels and may inadvertently merge signals from functionally disparate regions. By eliminating the necessity for spatial filtering, the Functionnectome offers a pathway to conduct statistical analyses with a heightened degree of precision and sensitivity, thereby holding the potential to unveil the white matter circuits that underpin brain interactions. We employed the Functionnectome in combination with three distinct fMRI paradigms that disentangle attention from conscious perception (Chica et al., 2013, 2016; Martín-Signes, Paz-Alonso, et al., 2019). Specifically, alerting, spatial orienting, or executive attention were manipulated in each paradigm by presenting an alerting tone, a peripheral cue, and a Stroop task, respectively for each process. The target was a near-threshold Gabor stimulus individually titrated to be perceived approximately 50% of the time. Participants had to discriminate the orientation of the Gabor's lines (except in the executive attention task), and to detect and report whether or not they saw the target (see methods section for a detailed description of the tasks). Through a conjunction analysis of the contrast Seen > unseen, we aimed to reveal specific networks that underlie visual consciousness, expanding our understanding of the neural correlates of visual consciousness independently from attention.

Added to the Methods (pages 12-13, lines 291-299): In the three experiments, participants underwent a titration procedure that ensured that 50% would be seen trials (i.e., participants reported that they consciously perceived the target and correctly located it), while the other 50% were unseen (i.e., participants indicated that they did not perceive the target). Indeed, when the target was consciously detected, the discrimination response (i.e., identifying the orientation of the Gabor's lines) was above chance, while it was at chance when it was not detected. Additionally, the percentage of false alarms (i.e., reporting to detect the target when it was not presented) was kept very low. Please, see also footnotes 1 and 2 (pages 12 and 13).

Comment 2: The results are over-interpreted. For instance, the authors claim to have demonstrated the pivotal role of integration in the conscious access to visual information, and many more examples of such over-interpretation can be found in the discussion. The

discussion should be revised so that it stays more closely to what the findings really are: additional evidence that the hypothesized white matter tracts are indeed important for conscious visual processing. Please also explain how the results could have been any different: couldn't we have known this in advance given the pattern of gray matter activations? It is not obvious how the analysis could, for instance, arbitrate between competing hypotheses of say one connecting white matter tract being involved but not another. After all, the Functionnectome does not measure involvement of white matter tracts but extrapolates from gray matter data, and the set of tracts that are highlighted are uniquely defined by the WM prior and GM data, which limits what types of questions can be answered with the method.

Thank you for your comment. We have now toned down our interpretations and also revised the discussion to highlight the importance of white matter tracts. We could not have known this in advance given the pattern of gray matter activations mainly because the circuitry supporting this set of activations could have been very different. For instance, frontal activation could have been supported by callosal exchange, which was not the case. We also now explain the Functionnectome methods in the introduction to clarify how it can arbitrate between one connecting white matter tract being involved but not another.

You can now read in the introduction (pages 2-3, lines 51-70): "The introduction of the Functionnectome represents a new methodology that integrates structural connectivity data within functional analysis. The Functionnectome takes the activity signals (BOLD time series) from the grey matter and combines them based on how these grey areas are connected to white matter areas. The strength of each connection influences the final signal. As a result, it creates a new type of 4D brain imaging data. This new data projects the brain activity from the grey matter onto the white matter, with the connections' strength influencing the outcome. This approach allows for a more nuanced examination of the interplay among various brain regions, moving beyond the traditional focus on their isolated functions in brain processes. Compared with earlier methods that initiated tractography directly from functional activation sites (Javad et al., 2014), the Functionnectome facilitates a data driven statistical analysis of the implicated white matter pathways. Moreover, it addresses a critical limitation of conventional functional fMRI analyses, which typically employ spatial filtering prior to statistical evaluation. Such traditional practices do not consider the structural connections between voxels and may inadvertently merge signals from functionally disparate regions. By eliminating the necessity for spatial filtering, the Functionnectome offers a pathway to conduct statistical analyses with a heightened degree of precision and sensitivity, thereby holding the potential to unveil the white matter circuits that underpin brain interactions."

Comment 3: It is not clear what the results presented in figure 2 and associated text are meant to convey. These results are introduced by saying that limited knowledge exists regarding which method (classical vs Functionnectome) better depicts brain functioning, but it is not clear by what standard. It appears that the authors define "better" in terms of the distribution of z-values, but it is not clear to me why one distribution would be better than another. The authors seem to conclude superior explanatory power by the Functionnectome analysis based on an increased number of positive z-values, but that is not valid because a better (i.e. true) model could very well have fewer activated voxels. Moreover, the number of positive z-values is expected to increase if the analysis selectively projects time-series into the white matter from a region of interest (mask M as defined in the original methods paper) with high z-values for the conjunction contrast (if this is not the case, please explain this better in the methods).

Thank you for your constructive critique. We acknowledge the oversight in our initial enthusiasm for the new method, and in response to your comments, we have carefully revised our manuscript. We now amended the text and removed any mention of superiority of one method compared to another. We do however now discuss more in detail the difference between the two methods as follows (page 10, lines 220-232):

"Our comparative analysis delineates the distinctions between traditional fMRI analyses and the Functionnectome approach in evaluating task-related fMRI signals. This examination reveals that, in contrast to classical analyses that primarily identify isolated activations, the incorporation of brain interactions along white matter pathways provides a significant alternative perspective. This perspective focuses on the exchanges between brain regions during visual conscious access. Importantly, our analysis detected much less deactivation than activation spreading along white matter pathways. This suggests that deactivation processes may not involve structural connections. Instead, they may be attributed to decreased activation or reduced oxygen consumption (Logothetis, 2008; Logothetis et al., 2001) in specific regions without direct structural links. Therefore, our findings propose that consciousness involves more than localized brain activation or deactivation, highlighting the importance of interactions between brain areas."

We apologize for the lack of clarity regarding how conjunction maps were calculated for the Functionnetome approach. For each methodological approach (i.e., classical or

Functionnetome), the conjunction map was calculated independently using the three task-derived contrast maps. Note that for the Functionnectome approach, the projection of the signal to the white matter was applied before task modeling and performing higher-level analysis. Therefore, this projection is independent of the z-values, as they are calculated in a later step. We have now clarified the method regarding conjunction map calculation as follows (page 16, lines 382-387): "A conjunction analysis using easythresh_conj tool from FSL (Nichols et al., 2005) was applied to the three seen > unseen statistical maps derived from the three experimental paradigms (phasic alerting, spatial orienting, and executive attention). Note that these conjunction maps were calculated independently using the contrast maps derived from each analysis approach."

Minor 1: Please insert explicit Results caption.

Thank you for noticing this omission.

Version 1:

Reviewer comments:

Reviewer #1

(Remarks to the Author)

The authors have addressed my suggestions, no further issues!

Reviewer #2

(Remarks to the Author)

all my concerns have been addressed

Reviewer #3

(Remarks to the Author)

Though the authors have adequately addressed most of my concerns, two remain:

In response to comment 1, the authors added the following lines of text to the manuscript:

"Moreover, it addresses a critical limitation of conventional functional fMRI analyses, which typically employ spatial filtering prior to statistical evaluation. Such traditional practices do not consider the structural connections between voxels and may inadvertently merge signals from functionally disparate regions. By eliminating the necessity for spatial filtering, the Functionnectome offers a pathway to conduct statistical analyses with a heightened degree of precision and sensitivity, thereby holding the potential to unveil the white matter circuits that underpin brain interactions".

I am unclear as to why spatial filtering would be a critical limitation of conventional fMRI analyses that does apply to the Functionnectome analysis. How does the latter eliminate the necessity for spatial filtering? The problem of merging signals from functionally disparate regions does not go away by not smoothing the data (because MRI data is intrinsically smooth and fMRI responses involved a point spread) and there are multiple valid reasons for spatial filtering (e.g. SNR improvement i.r.t. matched filter theorem, spatial smoothing to address anatomical differences between subjects, application of Gaussian Random Field theory) that also seem to apply to the Functionnectome analysis (or at least the first part of it). Please either clarify or remove this part of the text.

In response to comment 2, the authors state:

"We could not have known this in advance given the pattern of gray matter activations mainly because the circuitry supporting this set of activations could have been very different. For instance, frontal activation could have been supported by callosal exchange, which was not the case".

I understand that some white matter pathways do, and others do not get "flood-filled" with the fMRI activations of a reference voxel. However, the way I understand the formulas is that these differences are the result of that reference voxel having a fixed set of streamlines with different strengths (forming the priors). That is, in the example given, the streamlines seeding from the reference voxel simply didn't include callosal fibres very often, so these had smaller weights. Because the functional connectivity that is projected into the WM is the result of a simple multiplication of the GM activation and fixed WM priors, we shouldn't be very surprised about the results. Thus, contrary to the author's assertion, the frontal activation could not have been supported by callosal exchange, and we know this in advance because the streamlines of the reference voxel fully determine that the activity flows into some GM structures and not others.

If so, an important corollary is that the statistical testing on the WM "activation" would be nonsensical because (1) there is no level of surprise to quantify, and (2) the WM activation is not even a measurement and similar result could be obtained by simply projecting the GM z-scores into the WM (which would be statistically more kosher because the current way of

performing a mass univariate test on the WM voxels would not be accounting for all of the voxel timeseries being a scaled version of a single GM activation pattern).

Perhaps I misunderstand what's going on, but the given explanation does not convince me that the method is more than just a convenient tool for visualising what WM tracts are involved in certain tasks with questionable statistical inference step tagged on. Visualisation is not a bad thing per se and can lead to new insights, but if that is all there is to it, the authors should not present it as a tool for statistical inference.

Author Rebuttal letter:

Reviewer #1

The authors have addressed my suggestions, no further issues!

We thank the reviewer for their invested time and constructive comments, which have helped to improve the quality of the manuscript.

Reviewer #2

All my concerns have been addressed

We thank the reviewer for their invested time and constructive comments, which have helped to improve the quality of the manuscript.

Reviewer #3

Though the authors have adequately addressed most of my concerns, two remain:

Comment 1: In response to comment 1, the authors added the following lines of text to the manuscript:

“Moreover, it addresses a critical limitation of conventional functional fMRI analyses, which typically employ spatial filtering prior to statistical evaluation. Such traditional practices do not consider the structural connections between voxels and may inadvertently merge signals from functionally disparate regions. By eliminating the necessity for spatial filtering, the Functionnectome offers a pathway to conduct statistical analyses with a heightened degree of precision and sensitivity, thereby holding the potential to unveil the white matter circuits that underpin brain interactions”.

I am unclear as to why spatial filtering would be a critical limitation of conventional fMRI analyses that does apply to the Functionnectome analysis. How does the latter eliminate the necessity for spatial filtering? The problem of merging signals from functionally disparate regions does not go away by not smoothing the data (because MRI data is intrinsically smooth and fMRI responses involved a point spread) and there are multiple valid reasons for spatial filtering (e.g. SNR improvement i.r.t. matched filter theorem, spatial smoothing to address anatomical differences between subjects, application of Gaussian Random Field theory) that also seem to apply to the Functionnectome analysis (or at least the first part of it). Please either clarify or remove this part of the text.

Thank you for your comment. As the reviewer mentioned, among other reasons, smoothing aims at improving the signal-to-noise ratio (SNR) using a weighted average of the local signal from neighboring voxels. However, as the Functionnectome method combines the signal from distant yet structurally linked voxels, this has an analogous effect of improving the SNR, eliminating the necessity for additional spatial filtering. Therefore, the difference with traditional spatial smoothing is that in the Functionnectome method, this is guided by actual brain circuits. However, we agree with the reviewer that, at this stage, we cannot assume that one approach (i.e., merging the signal from structurally linked voxels) is more appropriate or has fewer limitations than the other (i.e., merging the signal from neighboring voxels). Following the reviewer's comment, we have now revised this part of the text and removed the reference to spatial smoothing as a limitation of traditional fMRI analysis (see pages 2-3, lines 51-66).

Comment 2: In response to comment 2, the authors state:

“We could not have known this in advance given the pattern of gray matter activations mainly because the circuitry supporting this set of activations could have been very different. For instance, frontal activation could have been supported by callosal exchange, which was not

the case.

I understand that some white matter pathways do, and others do not get 'flood-filled' with the fMRI activations of a reference voxel. However, the way I understand the formulas is that these differences are the result of that reference voxel having a fixed set of streamlines with different strengths (forming the priors). That is, in the example given, the streamlines seeding from the reference voxel simply didn't include callosal fibres very often, so these had smaller weights. Because the functional connectivity that is projected into the WM is the result of a simple multiplication of the GM activation and fixed WM priors, we shouldn't be very surprised about the results. Thus, contrary to the author's assertion, the frontal activation could not have been supported by callosal exchange, and we know this in advance because the streamlines of the reference voxel fully determine that the activity flows into some GM structures and not others.

If so, an important corollary is that the statistical testing on the WM 'activation' would be nonsensical because (1) there is no level of surprise to quantify, and (2) the WM activation is not even a measurement and similar result could be obtained by simply projecting the GM z-scores into the WM (which would be statistically more kosher because the current way of performing a mass univariate test on the WM voxels would not be accounting for all of the voxel timeseries being a scaled version of a single GM activation pattern).

Perhaps I misunderstand what's going on, but the given explanation does not convince me that the method is more than just a convenient tool for visualising what WM tracts are involved in certain tasks with questionable statistical inference step tagged on. Visualisation is not a bad thing per se and can lead to new insights, but if that is all there is to it, the authors should not present it as a tool for statistical inference.

Thank you for your detailed feedback and for engaging critically with our manuscript. We appreciate the opportunity to clarify and discuss our methodology and its implications. While this manuscript focuses on the application of our method to visual consciousness rather than introducing a new method, we are glad to elaborate on its advantages and limitations during this review process.

Our approach draws inspiration from the disconnectome (Foulon et al. 2018), which converts brain lesions into disconnection maps, and aims to project the BOLD signal onto the white matter. Our goal is to facilitate the exploration of the functional role of the white matter. This method has shown promise in various domains, including motor, language, and working memory functions (Nozais et al. 2021), and exploring white matter's support for functional connectivity (Nozais et al. 2023).

We respectfully disagree with your assertion that our method can be reduced to "simply projecting the GM z-scores into the WM." By performing a weighted average of the BOLD signal based on the probability of connection in the white matter and then conducting a standard general linear model (GLM), we are assessing whether the involvement of specific white matter tracts is significant. In other words, the Functionnectome increases the SNR of structurally related functional signals. Therefore, the Functionnectome priors can be considered anatomically-informed smoothing kernels that aim to maximize relevant functional signals across structural-functional circuits. This approach goes beyond merely tracking from significant functional activations in the grey matter.

We have now expanded the explanation of the method in the introduction, and we hope this will help readers address concerns similar to those raised by the reviewer. We have also reviewed the manuscript to remove any reference to terminology such as 'white matter activation', which we acknowledge is a very different measure. We hope this clarifies our methodology and addresses your concerns.

Added to the introduction (pages 2-3, lines 51-66): 'The introduction of the Functionnectome represents a new methodology that integrates structural connectivity data within functional analysis. The Functionnectome takes the activity signals (BOLD time series) from the grey matter and combines them based on how these grey areas are connected to white matter areas. The strength of each connection influences the final signal. As a result, it creates a new set of 4D brain imaging data. This new data projects the brain activity from the grey matter onto the white matter, with the connections' strength influencing the outcome. By performing a weighted average of the BOLD signal based on the probability of connection in the white matter and then conducting a standard general linear model (GLM), we are assessing whether the involvement of specific white matter tracts is significant. This approach allows for a more nuanced examination of the interplay among various brain regions, moving beyond the traditional focus on their isolated functions in brain processes. Compared with earlier methods that initiated tractography directly from functional activation sites, the Functionnectome facilitates a data

driven statistical analysis of the implicated white matter pathways.â

References

Foulon C, Cerliani L, KinkingnÃ©hun S, Levy R, Rosso C, Urbanski M, Volle E, Thiebaut de Schotten M. Advanced lesion symptom mapping analyses and implementation as BCBtoolkit. *Gigascience*. 2018 Mar 1;7(3):1-17. doi: 10.1093/gigascience/giy004. PMID: 29432527; PMCID: PMC5863218.

Nozais V, Forkel SJ, Foulon C, Petit L, Thiebaut de Schotten M. Functionnectome as a framework to analyse the contribution of brain circuits to fMRI. *Commun Biol*. 2021 Sep 2;4(1):1035. doi: 10.1038/s42003-021-02530-2. PMID: 34475518; PMCID: PMC8413369.

Nozais V, Theaud G, Descoteaux M, Thiebaut de Schotten M, Petit L. Improved Functionnectome by dissociating the contributions of white matter fiber classes to functional activation. *Brain Struct Funct*. 2023 Dec;228(9):2165-2177. doi: 10.1007/s00429-023-02714-y. Epub 2023 Oct 7. PMID: 37804431.

Version 2:

Reviewer comments:

Reviewer #3

(Remarks to the Author)

Unfortunately, I remain unconvinced about the validity of the Functionnectome analysis and associated statistical inference. The authors basically repeat the same general answer about what the analysis does, while they could instead provide a more formal analytical demonstration that my "assertion" would be incorrect. That said, I do not contest the insights per se, but rather the lack of critical reflection on the (possible) limitations of the applied method. As further discussion seems pointless without more formal analytical support of the claims made, I leave it to the discretion of the editors whether it is necessary to better address this concern or not.
